# Electromagnetic Wave Absorption Properties of Structural Conductive ABS Fabricated by Fused Deposition Modeling

**DOI:** 10.3390/polym12061217

**Published:** 2020-05-27

**Authors:** Wenwen Lai, Yan Wang, Junkun He

**Affiliations:** School of Materials Science and Engineering, Wuhan Institute of Technology, Wuhan 430074, Hubei, China; lightwee@163.com (W.L.); mrhejk@163.com (J.H.)

**Keywords:** microstructures, honeycomb, wood-pile, FDM, electromagnetic wave absorption

## Abstract

To obtain excellent electromagnetic wave (EMW) absorption materials, the design of microstructures has been considered as an effective method to adjust EMW absorption performance. Owing to its inherent capability of effectively fabricating materials with complex various structures, three-dimensional (3D) printing technology has been regarded as a powerful tool to design EMW absorbers with plentiful microstructures for the adjustment of EMW absorption performance. In this work, five samples with various microstructures were prepared via fused deposition modeling (FDM). An analysis method combining theoretical simulation calculations with experimental measurements was adopted to investigate EMW absorbing properties of all samples. The wood-pile-structural sample possessed wider effective absorption bandwidth (EAB; reflection loss (RL) <−10 dB, for over 90% microwave absorption) of 5.43 GHz and generated more absorption bands (C-band and Ku-band) as compared to the honeycomb-structural sample at the same thickness. Designing various microstructures via FDM proved to be a convenient and feasible method to fabricate absorbers with tunable EMW absorption properties, which provides a novel path for the preparation of EMW absorption materials with wider EAB and lower RL.

## 1. Introduction

In recent years, electromagnetic wave (EMW) radiation has drawn deep attention due to its great damage to wireless communication and the health of human beings. Therefore, the development of radar absorbing materials (RAMs) with high EMW absorbing performance has become increasingly urgent in military and civilian applications [1,2,3,4,5]. Although massive research projects on EMW absorption materials have proceeded, designing optimized EMW absorbers with a lower reflection loss (RL), a smaller thickness, and a wider effective absorption bandwidth (EAB) remains a serious challenge [6,7,8]. Nowadays, plenty of polymeric nanocomposite RAMs have been prepared by adding one or more types of absorbing agents to the polymer matrix [9,10,11]. In addition, nanocomposite RAMs have proven to be an effective material to adjust the EMW absorption performance by designing the composition and microstructures of the composite [12,13,14]. Moreover, recent research indicated that structural EMW absorbers have significantly developed modern stealth materials in their capacity to regulate the macrostructure of absorbers, which play an important role in adjusting the impedance match of EMW absorbers [15,16,17]. Honeycomb and its composite structures have attracted tremendous attention because of their significant benefits on energy absorption, thermal buckling resistance, and electromagnetic absorption properties. The height, maximum inner radius, and thickness of the honeycomb are considered the main factors affecting RAM properties [18,19,20,21]. Sandwich structures with honeycomb cores have also been widely investigated and applied as engineering structures for their excellent mechanical and EMW absorption performance [22,23].

Multilayer EMW absorbing materials with a gradient index (GRIN) have also demonstrated effectiveness in improving the EMW absorption performance. Attributed to a better impedance matching, the EMW can propagate deeper into the absorber, which considerably reduces the reflection of EMW on the surface of the material [24,25,26,27]. However, most multilayer absorbing materials are only spliced from as-prepared single-layer materials using some adhesives, such as glue, thereby generating an absorber with poor mechanical properties. In addition, the structural design of single-layer absorbing materials prepared by traditional manufacturing systems has also been limited.

Three-dimensional (3D) printing technology has attracted enormous attention as compared to traditional manufacturing processes due to its inherent capability of printing unique and complex geometries using computer-aided design [28,29,30,31,32]. In addition, printed polymeric composites have excellent and more controllable properties due to the plethora of reported research projects on their printing parameters and methods [33,34,35]. As such, 3D printing technology is able to provide a more efficient research scheme to properly design the single-layer absorbing layer structure to allow deeper EMW propagation inside the materials and for effective dissipation.

In this work, fused deposition modeling (FDM) was employed to fabricate conductive acrylonitrile butadiene styrene (ABS) samples with various microstructures. The electromagnetic parameters of conductive ABS were measured. The EMW absorbing performance of all structural conductive ABS samples was investigated by theoretical simulation calculations and experimental measurements. In addition, the effects of the microstructures on the EMW absorbing properties of the absorbers were discussed.

## 2. Experimental Setup

### 2.1. Materials

The conductive ABS pellets were prepared in our laboratory. The basic thermo–mechanical properties and conductivity of the conductive ABS pellets are presented in Table 1.

### 2.2. Preparation of the Conductive ABS Filaments

The preparation of conductive ABS filaments was carried out in a single-screw extruder (TP-07, Dongguan Songhu Plastic Machinery Co., Ltd., Dongguan, China). The prepared conductive ABS pellets were first dried in a vacuum oven at 80 °C for 4 h to prevent the generation of bubbles during the extrusion process. As shown in Figure 1a, the first, second, and third temperature zones of the single-screw extruder were set to be 180, 215, and 205 °C, respectively. The temperature of cooling water was set to be 60 °C. The screw and tractor rotation speeds were 980 and 450 rpm, respectively. The dried conductive ABS pellets were fed into the feed inlet in batches. By adjusting screw and tractor rotation speeds, the conductive ABS filaments (diameter of 2.85 mm) were obtained.

### 2.3. The 3D Printing Process

The extruded conductive ABS filaments were printed using an FDM 3D printer (Ultimaker 3, Ultimaker, Utrecht, The Netherlands) under the settings given in Table 2 to obtain the desired print geometries. The typical printing process is presented in Figure 1b. The prepared monofilament (diameter of 2.85 mm) was fed into the printer via a pinch roller. The nozzle with a diameter of 0.4 mm was used to melt the fed filament under a temperature of 260 °C. The temperature of the platform was set at 100 °C, and the adhesion sheets were used in the build platform to ensure the excellent adhesive attraction between printed objects and the platform. A slow print speed of 30 mm/s and a small layer height of 0.1 mm were set to obtain the more exquisite printed objects. In addition, the infill density was set at 100% for a better mechanical property. The geometric modeling and corresponding printed final samples are displayed in Figure 2. For the electromagnetic parameter tests of conductive ABS, a concentric annular sample with an outer diameter of 7 mm, an inner diameter of 3.04 mm, and a thickness of 2 mm was printed. To investigate and test the EMW absorbing performance of the absorbers with various microstructures, samples #1, #2, #3, #4, and #5 with lengths and widths of 180 mm were printed. Among them, samples #1, #2, and #3 had the same structure with honeycomb pore side length (a) of 6 mm, and a honeycomb hole spacing (t) of 3 mm. However, the three samples held different thicknesses (h) of 1.5, 2.5, and 3.5 mm, respectively. Sample #4 maintained the same honeycomb hole spacing of 3 mm as compared to #1, #2, and #3, but with a smaller honeycomb pore side length of 3 mm and a thickness of 2.5 mm. Sample #5 with a wood-pile structure consisted of little wooden bars with dimensions of 180 mm × 5 mm × 1.2 mm. A distance of 5 mm was maintained between the little wooden bars.

### 2.4. Characterization and Testing

#### 2.4.1. Electrical Conductivity Testing

The resistance of conductive ABS filaments was obtained using a high resistance meter (GT-1864, Dongguan High-speed Rail Detection Instrument Co., Ltd., Dongguan, China), as shown in Figure 3. The electrical conductivity, σ, was calculated by the following equation:(1)$$\sigma =\frac{1}{\rho }=\frac{L}{RS},$$
where *R* is the measuring resistance, *L* is the length of filaments clamped by a high resistance meter, *S* is the cross-sectional area of the filaments, and ρ is the resistivity.

#### 2.4.2. Electromagnetic Parameters Testing

The electromagnetic parameters, complex relative permittivity ($\varepsilon_{r}=\varepsilon^{\prime }-j\varepsilon^{\prime \prime }$), and permeability ($\mu_{r}=\mu^{\prime }-j\mu^{\prime \prime }$) of conductive ABS, which was printed into a concentric annular shape, were measured by a vector network analyzer (N5224A, Keysight Technologies, Santa Rosa, CA, USA) in the range of 2–18 GHz.

#### 2.4.3. EMW Absorbing Performance Testing

On the basis of the arch method, the EMW absorbing performances of samples #1, #2, #3, #4, and #5 were evaluated by measuring the RL in the range 2–18 GHz using a vector network analyzer (N5224A, Keysight Technologies, Santa Rosa, CA, USA) according to the RAM testing standard of GJB 2038A-2011. All of the test samples were positioned on a metal plate in an anechoic chamber.

## 3. Results and Discussions

### 3.1. Theory of EMW Absorption

A series of electromagnetic waves propagated in the forward direction when a beam of unit amplitude electromagnetic wave was placed perpendicularly on a multilayer absorber. In addition, reflected waves propagated in the reverse direction were generated in the absorption layer, as shown in Figure 4.

According to Kraus’s transmission line impedance conversion equation [36], the wave impedance, Zi, of the *i*-th layer can be calculated as follows:(2)$$Z_{i}=\eta_{i}\frac{Z_{i-1}+\eta_{i}\tan h\left(\gamma_{i}d_{i}\right)}{\eta_{i}+Z_{i-1}\tan h\left(\gamma_{i}d_{i}\right)},$$
where $Z_{i-1}$ is the wave impedance at the interface between the *i*-th layer and the *i*-1 layer.

The parametric intrinsic impedance, $\eta_{i}$, is given as
(3)$$\eta_{i}=\eta_{0}\sqrt{\frac{\mu_{ri}}{\varepsilon_{ri}}}$$
(4)$$\eta_{0}=\sqrt{\frac{\mu_{0}}{\varepsilon_{0}}}=\sqrt{\frac{1}{1}}=1,$$
where $\eta_{0}$, $\mu_{0},$ and $\varepsilon_{0}$ represent the characteristic impedance, permeability, and permittivity of free space, respectively.

The complex propagation constant of the *i*-th layer was calculated by the following equation:(5)$$\gamma_{i}=\frac{j2\pi f}{c}\sqrt{\mu_{ri}\varepsilon_{ri}},$$
where *c* is the velocity of light, *f* is the frequency of incident electromagnetic waves, and *j* is an imaginary unit.

The complex relative permeability, $\mu_{ri}$, and complex relative permittivity, $\varepsilon_{ri}$, of the *i*-th layer were calculated by the following equations:(6)$$\mu_{ri}={\mu_{ri}}^{\prime }-j{\mu_{ri}}^{\prime \prime }$$
(7)$$\varepsilon_{ri}={\varepsilon_{ri}}^{\prime }-j{\varepsilon_{ri}}^{\prime \prime }.$$

*Z*_0_ of the metal plate on the surface of the first layer was equal to zero; thus, $Z_{i}$ can be obtained according to Equation (2), 1≤i≤n.

Therefore, the reflection loss, RL, at the interface between air and multilayer absorbing material can be obtained using the equation as follows:(8)$$RL=20lg\left|\frac{Z_{n}-\eta_{0}}{Z_{n}+\eta_{0}}\right|.$$

### 3.2. Electromagnetic Property Analyses

The complex relative permittivity ($\varepsilon_{r}=\varepsilon^{\prime }-j\varepsilon^{\prime \prime }$) and permeability ($\mu_{r}=\mu^{\prime }-j\mu^{\prime \prime }$) both consisted of the real ($\varepsilon^{\prime }$ and $\mu^{\prime }$) and imaginary ($\varepsilon^{\prime }$ and $\mu^{\prime \prime }$) parts. Among them, $\varepsilon^{\prime }$ and $\mu^{\prime }$ determined the storage capability of electric and magnetic energy, respectively, while $\varepsilon^{\prime \prime }$ and $\mu^{\prime \prime }$ determined the deterioration capability of electric and magnetic energy, respectively [37]. Conventionally, the dielectric loss tangent $\left({\tan\delta }_{E}=\varepsilon^{\prime \prime }/\varepsilon^{\prime }\right)$ and magnetic loss tangent $({\tan\delta }_{M}=\mu^{\prime \prime }/\mu^{\prime }$) were used to describe electromagnetic loss capacity [38]. For an ideal absorber, the equation $\varepsilon_{r}=\mu_{r}$ in a wide frequency range was regarded as a necessity, and meanwhile, ${\tan\delta }_{E}$ and ${\tan\delta }_{M}$ were also regarded to be immensely large [39]. Obviously, electromagnetic parameters $\varepsilon_{r}$ and $\mu_{r}$ of the materials played a substantial role in the EMW absorbing performance of absorbers.

For non-magnetic materials, the $\mu^{\prime }$ and $\mu^{\prime \prime }$ values of the complex relative permeability were 1 and 0, respectively [40]. As shown in Figure 5c,d, the values of $\mu^{\prime }$, $\mu^{\prime \prime },{\;\mathrm{and}\;\tan\delta }_{M}$ varied in the range of 0.95–1.05, 0–0.1, and 0–0.12, respectively, indicating the RL values were mainly determined by the complex relative permittivity $\varepsilon_{r.}$ The measured frequency dependence of $\varepsilon^{\prime }$ and $\varepsilon^{\prime \prime }$ is shown in Figure 5a. As shown in Figure 5a, the values of $\varepsilon^{\prime }$ were in the range of 7.23–19.49 and gradually decreased with the increase of frequency, indicating a typical frequency dispersion behavior induced by polarization lag in high frequency [40]. The ε″ values also varied inversely with frequency, fluctuating from 24.21 to 7.44 with the variation of the frequency. It was worth noting that the values of $\varepsilon^{\prime \prime }$ were slightly larger than those of $\varepsilon^{\prime }$ in the low-frequency region, such that a higher dielectric loss tangent was generated. As demonstrated in Figure 5b, in the low-frequency range of 2–4 GHz, larger ${\tan\delta }_{E}$ values ranging from 1.06 to 1.24 were observed, while in the frequency range of 4–18 GHz, the values of ${\tan\delta }_{E}$ fluctuated from 0.8 to 1. Attributed to the superior conductivity of $6.3\times 10^{-3}$ S/cm, conductive ABS possessed an overall larger dielectric loss tangent ${\tan\delta }_{E}$, presenting enormous capability to dissipate electromagnetic energy.

### 3.3. EMW Absorbing Performance Analyses

According to the transmission-line theory and Equations (2)–(8), RL of single-layer absorbers could be simplified as
(9)$$Z_{1}=\eta_{0}\sqrt{\frac{\mu_{r1}}{\varepsilon_{r1}}}\tan h\left(\frac{j2\pi fd_{1}}{c}\sqrt{\mu_{r1}\varepsilon_{r1}}\right)$$
(10)$$RL=20lg\left|\frac{Z_{1}-\eta_{0}}{Z_{1}+\eta_{0}}\right|.$$

Based on the complex relative permittivity and permeability depicted in Figure 5, the RL of single-layer samples was calculated by Equations (9) and (10). As a result, Figure 6a,b shows the frequency dependence of the RL of single-layer samples when the thickness increased from 0.5 to 4 mm in the frequency range of 2–18 GHz. It can be clearly seen that the single-layer conductive ABS samples presented only a minimum RL of −10.05 dB at a corresponding thickness of 1.5 mm, indicating more than 90% EMW absorption. Other than that, all of the samples with other thicknesses showed an EAB of 0. Obviously, single-layer conductive ABS samples without consideration of the microstructures design exhibited difficulty in meeting the demands for use as RAMs.

Figure 6c shows the experimental curves of RL versus frequency of samples #1, #2, and #3, which were printed with the same honeycomb structure systems but hold different thicknesses of 1.5, 2.5, and 3.5 mm, respectively. As can be seen in Figure 6c and Table 3, sample #1 with a thickness of 1.5 mm presented the best EAB value of 4.97 GHz among the three samples, covering most of the Ku-band (12–18 GHz). Sample #2 with a thickness of 2.5 mm showed the lowest minimum RL (RL-min) of −23.37 dB, had an EAB value of 3.86 GHz, and covered almost the whole X-band (8–12 GHz). With the largest thickness of 3.5 mm, sample #3 showed the highest RL-min of −16.94 dB and narrowest EAB of 2.13 GHz, indicating the weaker EMW absorbing performance as compared to samples #1 and #2.

In addition, it can be clearly seen in Figure 6c that the RL absorption peaks shifted to lower frequencies with increasing thickness. The phenomenon can be verified by the quarter-wavelength (λ/4) attenuation model [41]. For λ/4 cancellation, the relationship between the thickness of absorber, $d_{m}$, and the corresponding frequency at the RL absorption peak, $f_{m}$, is given by the following equation:(11)$$d_{m}=\frac{nc}{4f_{m}\sqrt{\left|\mu_{r}\varepsilon_{r}\right|}}\;(n=1,\;3,\;5,\;\ldots ),$$
where $\mu_{r}$ is the complex relative permeability, $\varepsilon_{r}$ is the complex relative permittivity, and *c* is the velocity of light in the free space. As a result, Figure 6d presents the frequency dependence of the calculated λ/4 thickness and the corresponding fit of samples #1, #2, and #3. $d_{m}$ of samples #1, #2, and #3 clearly exhibited good fit with the λ/4 attenuation model.

What must be acknowledged is the design of the honeycomb structure considerably improved the EMW absorbing performance. A hypothetical microscopic mechanism of the electromagnetic wave energy dissipation of the honeycomb-structural specimens may explain this result, as shown in Figure 7. First, the hexagonal hole structure in the system increased the internal interfaces of the material, thus allowing the electromagnetic waves entering the system to provoke numerous reflection effects at these interfaces, thereby resulting in their gradual dissipation. Second, the EMW energy was further dissipated by the induced currents, which originated from the closed loop formed by the conductive fillers distributed along the hexagonal hole wall. In addition, many microcapacitor structures were formed between the conductive fillers and the insulated ABS substrate, which also attenuated a part of the EMW energy due to the hysteresis effect aroused from the interaction of the microcapacitors and incident electromagnetic waves [42]. At present, further experimental evidence is required to support the hypothetical microscopic mechanism.

To further investigate and compare the effect of diverse microstructure absorbers on the EMW absorbing performance, sample #4 with different cellular systems and sample #5 with a wood-pile structure were printed.

Sample #4 was printed with a denser honeycomb pore structure for a smaller hole side length of 3 mm as compared to #2. Figure 8a shows the experimental RL curves of samples #2 and #4 with the same thickness of 2.5 mm. The results are presented in Table 3. It can be clearly seen that when the honeycomb pore side length decreased from 6 to 3 mm, the RL-min decreased from −23.37 to −28.13 dB, and the EAB was broadened from 3.86 to 4.52 GHz. Definitely, sample #4 possessed better EMW absorbing performance. Interestingly, the EAB of both samples #2 and #4 covered almost the whole X-band (8–12 GHz). In general, enhancing the promotion of EMWs propagation into the material and reducing the reflection on the surface can improve the EMW absorption performance. Therefore, the improved absorbing performance of sample #4 may be attributed to its denser pore structure, which brought a larger number of internal interfaces, significantly reducing the reflection of EMWs on the material surface.

Figure 8b shows the experimental RL curves of sample #3 with a honeycomb structure and sample #5 with a wood-pile structure at the same thickness of 3.5 mm. Samples #3 and #5 exhibited different EMW absorbing performances ascribed to completely different microstructures, as shown in Figure 8b and Table 3. For sample #5, its RL-min value of −22.15 dB was lower than that of sample #3 at −16.94 dB. More importantly, the EAB of sample #5 covered 5.43 GHz, which was much wider than that of sample #3 at 2.13 GHz. In addition, the EAB of wood-pile-structural sample #5 mainly focused on the C-band (4–8 GHz) and Ku-band, whereas that of honeycomb-structural sample #3 only covered part of the C-band. The design of the wood-pile microstructure not only enhanced the EAB and RL-min of the absorber but also generated more absorption bands, exerting a significant influence on the EMW absorbing performance of the absorbers.

## 4. Conclusions

Single-layer conductive ABS absorbers with various microstructures were designed and fabricated via FDM in this work. The methods of theoretical simulation calculation and experimental measurement were adopted to investigate the effects of different microstructures on EMW absorbing properties of the samples.

The printed samples with a honeycomb structure system presented significantly improved EMW absorbing properties as compared to the single-layer conductive ABS samples that did not consider the microstructures. It presented the widest EAB (RL <−10 dB, for over 90% microwave absorption) of 4.97 GHz at a corresponding thickness of 1.5 mm and a honeycomb hole side length of 6 mm, covering most of the Ku-band (12–18 GHz).

A hypothetical microscopic mechanism of electromagnetic wave energy dissipation of the honeycomb-structural specimens may be helpful in explaining the improved EMW absorbing properties. First, the hexagonal hole structure in the system increased the internal interfaces of the material, thereby allowing electromagnetic waves entering the system to provoke numerous reflection effects at these interfaces, resulting in their gradual dissipation. Second, the EMW energy was further dissipated by the induced currents that originated from the closed loop formed by the conductive fillers distributed along the hexagonal hole wall. In addition, many microcapacitor structures were formed between the conductive fillers and the insulated ABS substrate, which also attenuated a part of the EMW energy due to the hysteresis effect aroused from the interaction of the microcapacitors and incident electromagnetic waves.

A wood-pile-structural absorber with a thickness of 3.5 mm was also successfully printed and, interestingly, presented an optimal EMW absorption performance. Not only was its EAB improved to 5.43 GHz, but it also showed the capability of generating more absorption bands. This work demonstrated that the design of microstructures exerted a significant influence on the EMW absorbing performance of absorbers. In addition, the 3D printing technique exhibited a remarkably convenient and effective method to fabricate an absorber with tunable EMW absorption properties, which provides great potential to further investigate RAMs with wider EAB and lower RL-min.

## Figures and Tables

**Figure 1 polymers-12-01217-f001:**
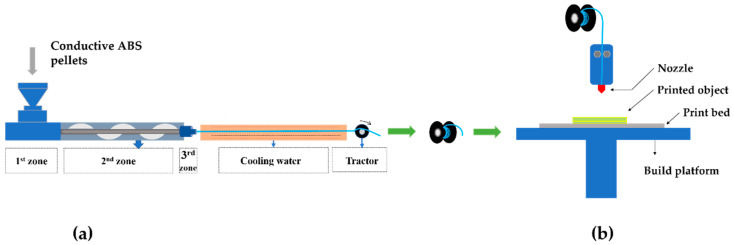
(**a**) Preparation of the conductive ABS filaments. (**b**) 3D printing process.

**Figure 2 polymers-12-01217-f002:**
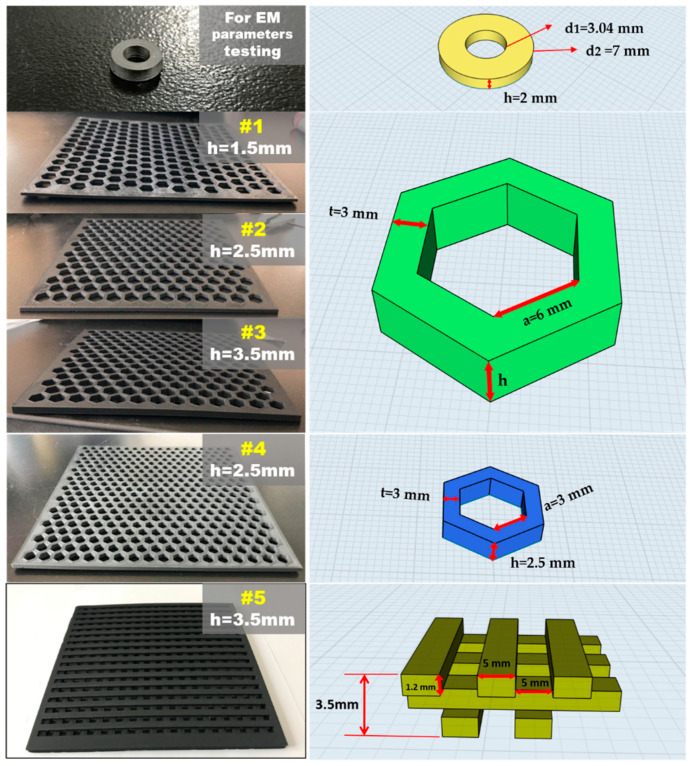
Geometric modeling and corresponding printed final samples by fused deposition modeling (FDM).

**Figure 3 polymers-12-01217-f003:**
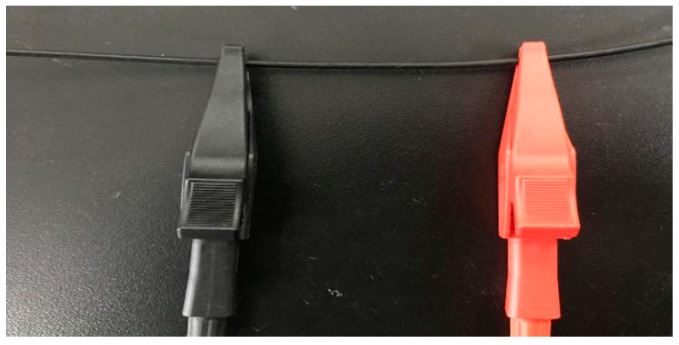
Measurement of the resistance of conductive ABS filaments.

**Figure 4 polymers-12-01217-f004:**
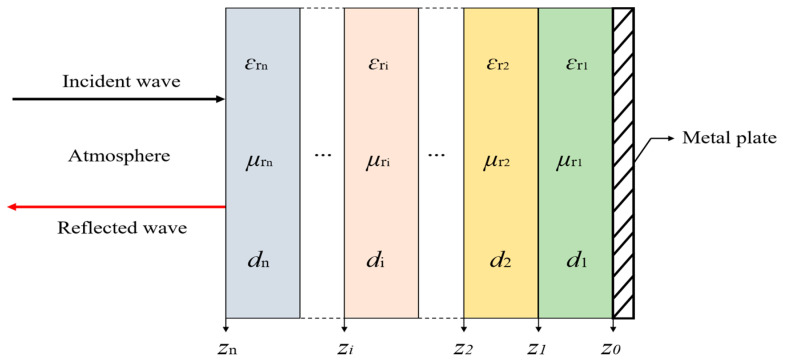
Schematic of a multilayer model with a normal incident wave.

**Figure 5 polymers-12-01217-f005:**
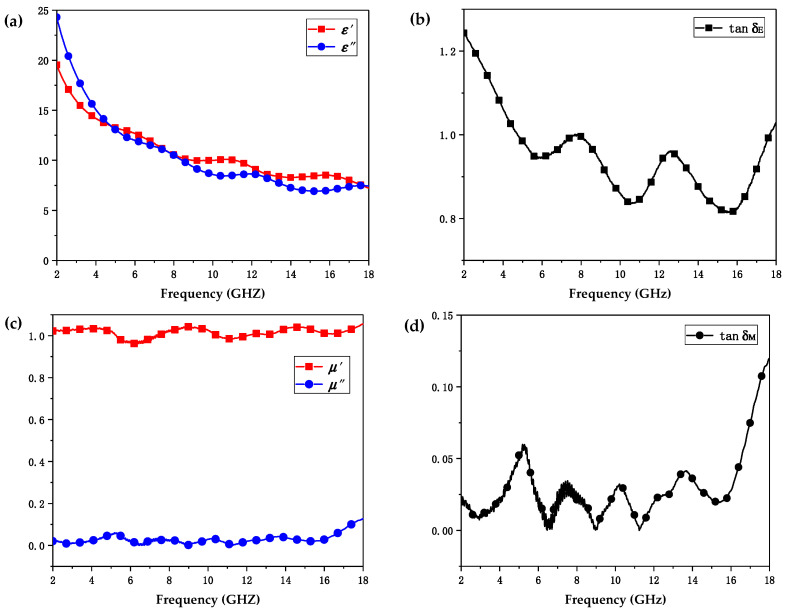
Electromagnetic property of conductive ABS. (**a**) Real parts ($\varepsilon^{\prime }$) and imaginary parts ($\varepsilon^{\prime \prime }$) of the complex relative permittivity. (**b**) The dielectric loss tangent ${\tan\delta }_{E}.$ (**c**) Real parts ($\mu^{\prime }$) and imaginary parts ($\mu^{\prime \prime }$) of the complex relative permeability. (**d**) The magnetic loss tangent ${\tan\delta }_{M}.$

**Figure 6 polymers-12-01217-f006:**
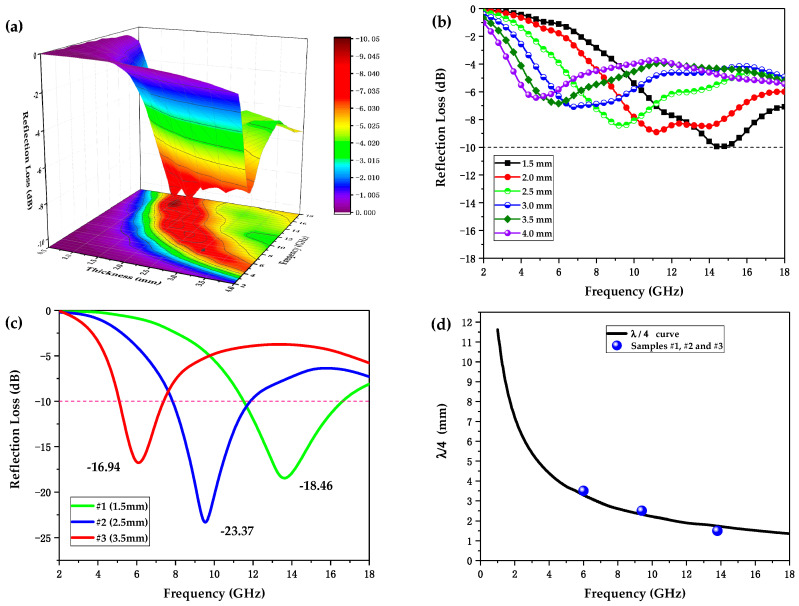
(**a**) 3D theoretically calculated reflection loss (RL) spectrums of single-layer conductive ABS samples when the thickness increases from 0.5 to 4 mm and (**b**) represents the corresponding 2D RL plots. (**c**) Experimental RL plots of samples #1, #2, and #3. (**d**) The λ/4 resonance model and the relation between absorber thickness and the corresponding frequencies at the RL absorption peaks.

**Figure 7 polymers-12-01217-f007:**
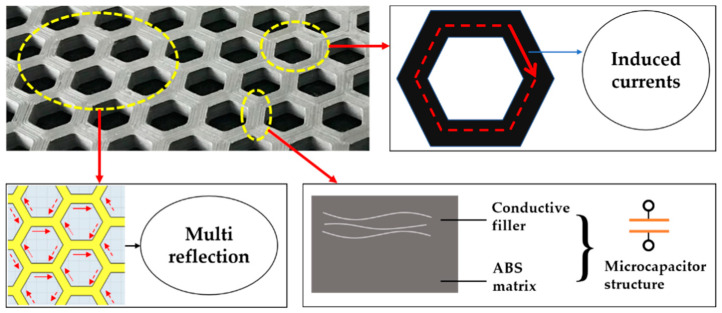
A hypothetical microscopic mechanism of the honeycomb-structural conductive ABS samples on electromagnetic wave (EMW) energy dissipation.

**Figure 8 polymers-12-01217-f008:**
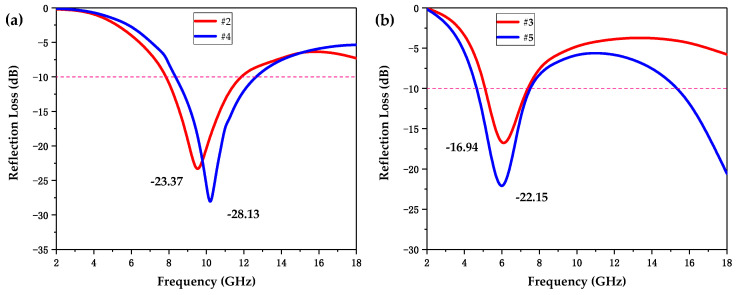
(**a**) Comparison of the experimental RL spectrums of samples #2 and #4 at the same thickness of 2.5 mm. (**b**) Comparison of the experimental RL spectrums of samples #3 and #5 at the same thickness of 3.5 mm.

**Table 1 polymers-12-01217-t001:** Basic thermo–mechanical and electrical properties of the conductive acrylonitrile butadiene styrene (ABS) pellets.

Properties	Test Methods	Unit	Values
Melt index	ASTM D1238 (200 °C, 5 KG)	g/10 min	6.1±0.3
Heat deflection temperature	ASTM D648	°C	90.0 ± 0.3
Tensile strength	ASTM D638	kg/cm^2^	505 ± 5
Impact strength	ASTM D256	J/m	162±2
Conductivity	Shown in 2.4.1	S/cm	$\left(6.3\pm 0.8\right)\times 10^{-3\;}$

**Table 2 polymers-12-01217-t002:** Settings of 3D printer.

Main Printing Parameters	Values
Layer height (mm)	0.1
Infill density (%)	100
Print speed (mm/s)	30
Nozzle temperature (°C)	260
Nozzle diameter (mm)	0.4
Build platform temperature (°C)	100

**Table 3 polymers-12-01217-t003:** Measured EMW absorbing properties of samples #1, #2, #3, #4, and #5.

Samples	#1	#2	#3	#4	#5
Apparent density (g/cm^3^)	0.49	0.53	0.51	0.56	0.57
EAB (RL < −10 dB) (GHz)	4.97	3.86	2.13	4.52	5.43
RL-min (dB)	−18.46	−23.37	−16.94	−28.13	−22.15

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
