# Peer review of "Electromagnetic Wave Absorption Properties of Structural Conductive ABS Fabricated by Fused Deposition Modeling"

_polymers, 2020, doi:10.3390/polym12061217_

Round 1

Reviewer 1 Report

The work by Lai et al presents an interesting approach to mitigate the impact of the EMW radiation, which is of interest nowadays due to the increasing amount of technologies that are based on such systems. Despite the interesting results, there are some issues to address prior to publication:

  1. The comparison in Figure is hard to convey because one is a 3D plot while the other is 2D. I believe the 3D plot is quite valuable but prevents a direct comparison. I suggest to move it to the supplementary section and make the corresponding 2D plots to have a sense of how close the model is to the experimental data. Moreover, some sort of goodness of fit approach should be used to make it quantitative. Finally, the collected data has no error bars and this is critical to really know if upon comparison the differences are statistically significant.
  2. I find that the idea for the mechanism of dissipation is quite interesting; however, I believe that the evidence is not enough to completely support it. I think that it should be only stated in the manuscript as a hypothetical idea that needs further experimental evidence. Moreover, this should be part of the conclusions.
  3. Figure 7 should also include the corresponding error bars and clearly state the number of replicas.
  4. Finally, the conclusions are quite specific and required more elaborate thinking to make them general and more easily digestible but a general audience.    

Reviewer 2 Report

This manuscript represents the characterization of 3D printed conductive ABS structures in terms of electromagnetic wave absorption behaviour. FFF technique was used to manufacture the 3D printed parts. The structure of the paper is well organized and the overall paper tells a logical story with a concrete conclusion. It is suitable for publication in the Polymers after minor revision.

1) Introduction and Reference sections should be improved with published works related to the analysis of 3D printed polymer  composites with improved properties, as for example:

[1] M.A. Caminero, J.M. Chacón, E. García-Plaza, P.J. Núñez, J.M. Reverte, J.P. Becar, Additive Manufacturing of PLA-Based Composites Using Fused Filament Fabrication: Effect of Graphene Nanoplatelet Reinforcement on Mechanical Properties, Dimensional Accuracy and Texture, Polymers 11(5) (2019) 799

2) Please include more details of the basic thermo-mechanical and electromagnetic properties of the conductive ABS pellets used in this study

3) Please explain in more detail the filament preparation and the 3D printing process

4) Please include the standard deviation of the results reported in this study

Round 2

Reviewer 1 Report

The authors have clarified and addressed my concerns regarding how data was analyzed and presented and clearly state that the mechanistic details require further experimental evidence.

There are a few minor grammar issues that I suggest should be addressed prior to publication.

Author Response

Dear reviewer:

I am very grateful for your comments about the manuscript. Our manuscript has been checked and corrected by native English-speaking editors. Attached is the language editing certificate.
